# Invited perspectives: How does climate change affect the risk of natural hazards? Challenges and step changes from the reinsurance perspective

Anja T. Rädler[1]

[1]Munich Reinsurance Company, Königinstraße 107, 80802 München

**Correspondence:** Anja T. Rädler (araedler@munichre.com)

Over the last decades, natural disasters have been causing ever increasing overall economic losses worldwide. For assessing these trends and the underlying risks, it is important to carefully study the main drivers behind such changes. One of these drivers is an increase in destructible wealth due to economic growth in many regions of the world, also combined with changes in the vulnerability of assets. Other drivers are changes in the frequency or intensity of natural hazards in the recent past and in the near future. In this contribution the focus will be on the latter driver.

As the author of this contribution is working with Munich RE, a globally engaged leading reinsurance company operating since 1880, the viewpoint here is a reinsurer's perspective. Insurance companies from markets all over the world are the customers of reinsurance companies. Reinsurers are covering the peak portion of their customers risk from natural hazards. They are capable of this role because their business model builds on a global diversification of those risks via the global setup of the company. Diversification is achieved by two ways: on the one hand by addressing different markets as for example USA, Europe and Australia which geographically have a low level of correlation in terms of natural catastrophes. On the other hand, by dealing with different types of risks which are ideally independent from a physical perspective, for example damage by hailstorms and earthquakes. Relevant natural catastrophe loss events are more or less globally distributed, for instance, in the year 2021, ranging from rainstorm Bernd in Europe in July to Hurricane Ida in the USA in August or to hailstorms in Australia in April. In addition to these meteorological events, also slowly evolving events such as droughts which enhances wildfire risk, cause substantial damages and losses. Primary insurers, which are often smaller and operate locally compared to reinsurers, could be hit so hard by a natural catastrophe that it affects their entire portfolio at once and they then do not have enough risk capital. Unlike reinsurers, they cannot diversify their risks with other risks in other regions.

For adequately dealing with these potential losses, a reinsurer has to quantitatively assess the individual risks of natural catastrophes and how these risks are changing over time with respect to climate change. Formally, risk is the product: probability of the hazard event and the consequence (i.e., the associated loss). Here it is important, that risk is not assessed from an individual "grid point" perspective, which is a common approach in scientific studies, but for an entire catastrophic event, i.e., for the total of all the assets / grid points in a region that might be affected by a regional catastrophe. Looking at individual grid points is a higly valuable approach and the basis for further analysis. For the reinsurance industry, the event definition is of major importance as the (overall) losses depend on aspects of spatial extent and duration.

As far as climate change is concerned, the challenges for the reinsurance industry are not necessarily the same as for the scientific community. While many scientific climate studies focus on changes of natural hazards at the end of the century, for reinsurance, the status quo and the near future (1-5 years) is most relevant. As reinsurance treaties are annually renewed, there is adaptive potential in the medium and long run. It is much more important to learn about changes in rare event intensities
and frequencies, that might be already underway and realize in the near-future temporal business horizon. In climate science, the prediction of the near future, seasonal or (multi-)decadal forecasts and their improvements are frequent contents of studies (Kirtman et al., 2013). Long-term effects and changes play a role in asset management (e.g. investments in forests which are growing for up to 80 years) and for future planning, e.g. to focus on certain markets and regions, or when addressing mitigation and adaptation to changing hazards. However, for the operative business the status quo is crucial and risk models are developed
to estimate, to the best of our ability, the risk associated with natural hazards. These risk models are subject to constant change and must therefore be updated regularly. At annual renewal, the treaty can and must be adjusted to reflect the actual risk. This is because risks of natural hazards can rapidly change regionally and also in their global scope due to short term effects such as internal climate variability. For the development of new future-proof risk models, the effects of climate change, e.g. changes in temperature or precipitation or storm intensities, which have already occurred in the past decades and may today be clearly
detectable, play an important role. This is because the large event sets of risk models are based on the historical events. Another component of risk, or rather exposure, is (critical) infrastructure, which can also be affected by climate change (e.g., riverways by low water levels or rail networks by record high temperatures).

In the fifth assessment report (AR5) of the Intergovernmental Panel on Climate Change (IPCC), Working Group II, the topic of risk was dealt with in a holistic way, which was a novelty in AR5 (IPCC, 2014). The sixth assessment report of the IPCC
(AR6) which assesses the physical science basis has been published early this year (IPCC, 2021), while the report of Working Group II that deals Impacts, Adaptation and Vulnerability will be published in 2022. Three components contribute to the risk as illustrated in Figure 1.

– First, the hazard, such as hurricane gust wind speeds, or hailstones of various diameters etc.

– Second, the exposure that is hit, which can be for example industrial, commercial or residential.

– Third, the vulnerability of the exposure that is hit. The more vulnerable the exposure, the higher the damage.

The second and third component of the risk, the exposure and vulnerability which describe the consequences, are changing constantly due to administrative standards (e.g., building codes), economic cycles and technological innovations. Climate change may have an indirect influence on these factors but this will not be addressed here. The first component of the risk, the hazard, can be directly influenced by natural variability and climate change, which plays an important role for model
development. For developing a risk model, the historic events of an individual hazard (e.g. gust wind speeds of tropical cyclones in the North Atlantic) are analyzed and a large event set that consists of $\gg 100,000$ years is stochastically derived from the historic data. If there are trends in the historic data due to climate change, this should be addressed, e.g. by a different weighting of the past decade relative earlier decades or by an explicit inclusion of such trends in the development of the event sets.

Otherwise, the risk assessment might be incorrect. Natural variability in the historic data should balance out in the stochastic
event set if the historic time span is long enough. However, for addressing the risk of the year to come, the hazard intensity
distribution can be adjusted on an annual basis given the phase of the natural variability. For instance, a positive phase of the
Indian Ocean Dipole in Australian spring will increase the bushfire risk relative to the long-term average.

In contrast to many scientific studies, which are primarily dedicated to answer the question of the impact of climate change
decades ahead or even towards the end of the century, the main focus in reinsurance is on operational business in the presence.
Hence, the most pressing scientific questions from the reinsurance point of view with respect to natural hazards are:

**What is the status quo of the individual hazards?**

Assessing the status quo of the risk of individual hazards is the foundation for the risk-adequate pricing of pending treaty
renewals. It also allows for the inference of low probability events and the scientifically controlled construction of worst-case
scenarios.

**How and where have past trends already altered hazard frequencies?**

Building hazard models relies on taking into account data from the past and it is essential to know how this data can be inter-
preted in the right way. Existing trends must be taken into account and cannot be assumed to balance out.


**If and how did climate change affect the individual hazards globally and regionally?**

Whether climate change affects the regional distribution and frequency, and severity of natural disasters must be answered for
individual hazards as well as their correlation across geographies.

**How can climate change be distinguished from natural variability?**

For many severe events controversial discussions have arisen whether they can be attributed to climate change or not. An-
swering this question is difficult and only the variation of distributions of magnitude and frequency can be examined. Natural
variability would balance out over a longer period of time while climate change might cause sustainable shifts. The World
Weather Attribution (WWA) initiative assesses to what extent single extreme weather events are influenced by climate change.
For example, Kreienkamp et al. (2021) found that the likelihood of an extreme rainfall event such as rainstorm Bernd in July
2021 has increased due to climate change.

**Is there a change in the distribution of individual hazards (e.g. at the extremes in high quantiles)?**

The mean values of the distributions play only a minor role in the risk assessment. A few extreme events from the tail of the
distribution are responsible for the majority of the losses. Hence, it is important to learn whether or not particularly frequen-
cies of rare and extreme event intensity levels have already changed due to climate change. Such analyses need support from
scientific climate modelling approaches. The challenge is great for the reinsurance industry because they have the pressure that
the extreme events of a changing tail of the distribution may already be happening, although it is not yet possible to quantify

them due to short time series, large uncertainties and a lack of research studies.


### Does climate change affect natural variability?

This may, for example, be the case for the El Niño-Southern-Oscillation (ENSO). As expected by some researchers, very extreme and rare levels of El Niño events may increase with climate change (Wang et al., 2019).


Answering the above questions is essential to adequately address the risk of natural hazards in the future. In order to be able to offer insurance for natural catastrophes, the accuracy of the models, i.e. their capability of assessing the risks are essential. Such tools have a significant influence on the functioning of the insurance system. Gross misjudgements can have serious consequences for the entire society.

In addition to answering the questions above, improvements in seasonal and decadal forecasting could be an important achievement by the scientific community.

Natural (or more precise internal) interannual climate variability can influence regional hazard activity and risk. On an annual or seasonal basis, risk or hazard activity can be enhanced or reduced by this variability. The mean state of the climate among other statistics are varying on all spatial and temporal scales, e.g. on a yearly basis, taking into account troposphere-stratosphere

or troposphere-ocean exchanges, or on a multidecadal basis, which involves ocean circulations. Examples of natural variability patterns are the Pacific Decadal Oscillation (PDO) and the El Niño-Southern-Oscillation (ENSO) in the Pacific, the Atlantic multidecadal variability/oscillation (AMV/AMO), the Atlantic meridional overturning circulation (AMOC) and the North Atlantic Oscillation (NAO) in the Atlantic, the Indian Ocean Dipole (IOD) in the Indian ocean. These natural variabilities can have a direct influence on extreme events in certain regions. One example is the decadal oscillation in hurricane (AMO) or

typhoon activity (PDO) that play a dominant role on the variability of losses caused by extreme events (Hoeppe, 2016). Further, with El Niño conditions, the hurricane frequency in the Atlantic Basin is reduced due to stronger vertical wind shear, and also higher atmospheric stability. The opposite phenomenon, La Niña enhances hurricane activity in that region (Gray, 1984; Bell and Chelliah, 2006). Another example of severe events are the bushfires in Australia in 2019/2020 which resulted from an exceptionally severe drought and high temperatures over parts of Australia enhanced by the positive phase of the IOD in

combination with an El Niño weather pattern. The 2019/2020 fire season in Australia produced record loss amounts of around US$ 2bn, of which US$ 1.6bn was insured (Munich Re, 2020). A study by van Oldenborgh et al. (2020) investigated changes in the risk of South-Australian bushfire weather due to anthropogenic climate change and found "a significant increase in the risk of fire weather as severe or worse as observed in 2019/20 by at least 30 %". Concerning wildfires in California, the annual extent of wildfire increased fifefold since the early 1970s, which is very likely driven by drying of fuels promoted due to

anthropogenic climate change (Williams et al., 2019).

Due to the high loss potential, increasing knowledge of how these natural variability phenomena and their interactions are affected by climate change is of interest for the reinsurance industry.

Improvements in seasonal forecasting are a focus of recent projects and first promising studies with achievements on the example of predicting heavy precipitation have been published (Hu et al., 2021). Large scale climate factors influence extreme precipitation which can lead to severe flooding events. Strong indicators in predicting precipitation are climate indices, which can be predicted at least several months in advance. The framework developed by Hu et al. (2021) can estimate flood economic loss risk several months in advance using climate factors. A precise forecast of flood risk plays an important role for adaptation and mitigarion efforts. However, it might also have an effect on the insurability of the flood risk as one precaution for insurability is that a loss event cannot be forseen in terms of its exact time and place.

Many of the questions raised above have been discussed in the scientific community for decades. The sixth assessment report of the IPCC (AR6) assesses the physical science basis and addresses the reliability of the knowledge about climate change based on the latest research (IPCC, 2021). In the AR6, the evidence has strengthened of observed changes in extremes such as heatwaves, heavy precipitation, floods, droughts, and tropical cyclones. For example, the AR6 concludes that an increase in the global proportion of major tropical cyclones (Category 3-5) over the last four decades is likely. This, and a poleward shift of the tracks (respectively the peak intensity) in the western North Pacific cannot be explained by natural variability alone. High confidence is estimated for the increases in heavy precipitation associated with tropical cyclones which is associated with human-induced climate change. Limitations are mentioned for an unambiguous and global detection of past trends (IPCC, 2021), which is of great interest for reinsurance companies, as described above.

A slightly different view of the risk issue in the context of climate change is provided by the Geneva Association, a leading international insurance industry think tank. In collaboration with its members and academic institutions, they analyze climate change risk in terms of both physical and transition risks for the insurance industry and describe the insurance industry's view of regulatory approaches to climate change risk (The Geneva Association, 2021a, b). Recent regulatory requirements implemented by the "Task Force for Climate Related Financial Disclosure" (TCFD) pose a major challenge for the reinsurance ecosystem and require intensive study in order to be able to substantiate a long-term strategy how to deal with investments sensitive to climate change. Ultimately the goal is to provide qualitative long-term projections up to 2050 and beyond. Public policies, regulations, technological development and economic cycles will directly and indirectly influence climate change over the next decades. Such socio-economic conditions represent multi-dimensional uncertainties which suggest quantitative considerations rather for shorter time frames and their combination with qualitative approaches for the long-run (The Geneva Association, 2021b).

Although there is a better understanding today of how climate change is affecting the frequency and severity of natural hazards, many questions especially concerning the risk are still open and are subject of further studies and projects.

One such project that focused on the risk component in Europe was the ARCS project (Analysis of changes in the Risk of Severe Convective Storms in Europe), which was funded by the Federal Ministry of Education and Research (BMBF) and a cooperation between the European Severe Storms Laboratory (ESSL) and Munich Re. In ARCS, it was analyzed whether trends of the frequency of convective hazards have been identified in the past 40 years in Europe (Rädler et al., 2018). Future changes in the occurrence of severe convective storms were projected using different climate scenarios (Rädler et al., 2019). In addition of addressing the hazard component, e.g. by identifying increases in hailstorm frequencies for both past and future

in Central and Eastern Europe, the hailrisk and the changing impact of hailstorms has been analyzed. Expected changes of the annual hail losses and largest hail diameter were identified based on large event sets applied to EuroCordex regional climate models. One of the main finding was, that a 100 year hailstorm loss could end up as a < 25 year event at the end of the 21st century in a RCP 8.5 climate scenario (ESSL, 2019).

Cooperation projects between the reinsurance industry and scientific institutions often have a different and extended perspective on extreme events. Scientists working in the industry also provide additional knowledge to the scientific community which is a win-win situation. Instead of identifying changes only on a grid point basis, the focus is often shifted towards the overall impact of natural hazard events at regional scale, presupposing a risk perspective including all three components, hazard, exposure and vulnerability.

*Author contributions.* This article was written by Anja T. Rädler.

*Competing interests.* No competing interests are present.

*Acknowledgements.* I would like thank my former colleague Dr. Eberhard Faust for the guidance and numerous discussions on risk, climate change and natural variability during the past seven years at Munich Re. I wish you all the best for your post MR time!

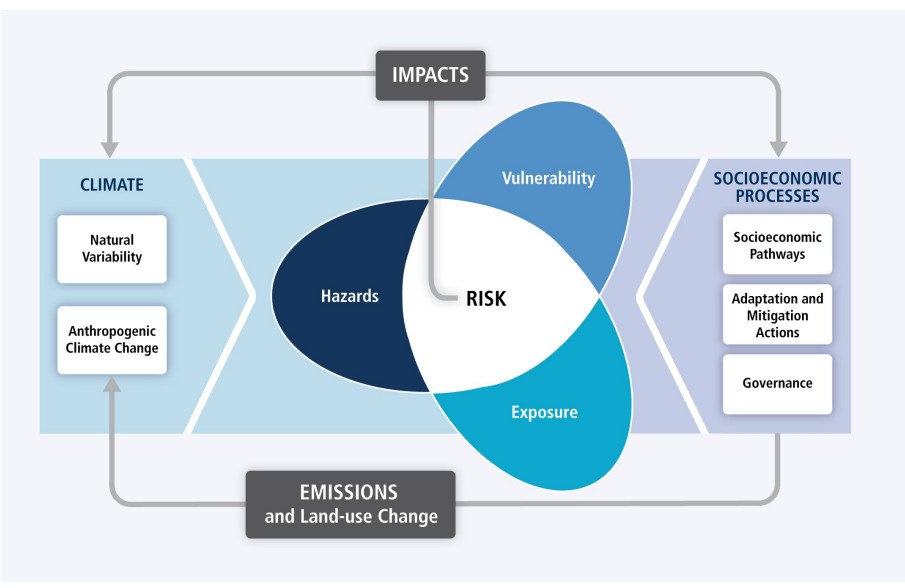

**Figure 1.** Illustration of the core concepts of the Working Group II of the fifth assessment report of the IPCC (IPCC, 2014).

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
