# Peer review of "Invited perspectives: How does climate change affect the risk of natural hazards? Challenges and step changes from the reinsurance perspective"

_Natural Hazards and Earth System Sciences, 2021_

## Author Response (AR1)

I would like to thank both reviewers for the in-depth comments, corrections and suggestions that have greatly improved the manuscript.

Please find my response below. I have answered each point individually (highlighted in bold and blue).

Reviewer 1:

Specific comments

L10-11: It is stated that there is a low correlation of severe natural hazards in geographically distant regions. Is there already enough scientific evidence for this assumption? I know an un-reviewed report carried out by the British Met Office and available as a file download on www.loyds.com (pdf-risk-reports-1781G-Lloyds-Met-Report-2016.pdf which came to this conclusion, but recent research on amplified Rossby waves or teleconnections modified by Arctic Amplification, for example, suggests that this may not be the whole truth.

Thank you for pointing this out. Indeed, severe natural hazards that occur in geographically distant regions might not be completely independent. Amplified Rossby waves or teleconnections modified by Arctic Amplification are two examples where research is currently focused on. Large scale natural variability phenomena such as the ENSO-Southern Oscillation are well known to have an effect on severe natural hazards in far distant regions.

My statement "… low level of correlation in terms of natural catastrophes" aimed to explain the business model of reinsurers. Reinsurers are only able to cover the peak risks of primary insurers if they act globally and thus achieve the greatest possible diversification. I have tried to illustrate this with the two examples of different markets and different types of risks. Different markets like USA, Europe and Australia can comprise possible correlations in individual perils, which are caused for example by ENSO. However, diversification is also achieved by different types of risks, such as hailstorms and earthquakes. Losses by a hailstorm in Europe and an earthquake in New Zealand have no correlation, as far as I know.

It is important for the reinsurers to know the individual risks of natural hazards, its possible correlations, and how these risks are changing over time with respect to climate change as best as possible to calculate adequate premiums and to adequately deal with the potential losses.

I have added a sentence to address your comment with the correlations.

L18-20: I assume that the intention of the author is not to criticize scientific work looking at events on an individual grid point basis, which could be understood from the wording. Thus, I suggest to change the wording, possibly pointing out the insurance industry's interest in additional aspects of extreme events, like their

spatial extent or duration. The existence of scientific research considering the (potential) loss arising from events should be mentioned. The background of why a "regional catastrophe" is of particular interest to the insurance industry, should be explained, as it is probably not immediately clear to all readers.

I did not want to criticize scientific work looking at events on individual grid point bases. This is highly valuable work and a basis for further analysis. However, I wanted to emphasize, that for the reinsurance industry, the definition of extreme events is of major importance. It all comes down to the losses caused by extreme events which depend on aspects of spatial extent and duration.

I modified the paragraph.

L22-25: The focus of climate studies on changes towards the end of the century is mostly explained by the need to find unambiguous results with respect to the greenhouse gas effects. Signals are weaker when considering the near future with lower GHG concentrations, and decadal climate variation is more important on the short time scale of 1-5 years. Research available on decadal climate prediction (and on multi-seasonal predictions could be mentioned/cited.

Yes. I have added citations for decadal climate prediction and multi-seasonal predictions.

L29: I assume this refers to risk models of the insurance industry. I do not understand why these models must be updated regularly, if they are working ok. Is this an effect of climate change, for example? Should updating refer to progress in seasonal to decadal predictions?

Risk models of the insurance industry are updated on a regular basis. One reason is to include the most recent events (e.g. the latest decade of hurricanes in the North Atlantic) in order to be able to compare upcoming events with historical event losses and to improve statistics when adding risks of the tail of extreme event distributions. If there is a consensus in science of the effect of climate change on certain hazards this can be addressed in a risk model update as well.

L67-68: While I understand that an unambiguous assignment of an event to climate change is formally impossible, there are several scientific studies on the role of climate change in making individual events more likely.

Yes, you are right. I will add a few citations.

L136-L150 I understand the benefits of the perspective establishing joint projects between the insurance industry and science. However, I do not understand why a detailed description of a particular project should be helpful. If the project is mentioned taken as an example in order to demonstrate the added value of insurance industry as a partner (mentioning it after the statement on joint projects),

the particular benefits should be explained. Of course, the question may come up why the insurance industry is not (individually or through some research agency) financing science on the open questions they regard particularly relevant. My guess is that this has a financial and research politics background, rather than a scientific reason.

The specific research project was mentioned to emphasize that such a project with scientists working in the insurance industry also provide additional knowledge which are of high interest for the broad scientific community. In addition the needs of the insurance industry to deal with changing risks was addressed. I tried to emphasize this win-win situation by giving concrete examples of the project and to motivate the community to continue with such cooperation projects.

Minor remarks

L 14 The relevance of mentioning the rainstorm Bernd and the Hurricane Ida (and some unnamed hailstorm in Australia) may not be obvious to all readers. You should provide a month/year of these events when clarifying your point.

Done

L16 I think a wildfire is not really a slowly evolving event like a drought. Actually, my understanding is that wildfires are generally related to drought.

Yes. I rephrased the sentence.

L32: I do not understand what is meant with "long-term effects from climate change that have already happened in the past decades. Please re-write.

Done

L36-38: Write as bullet points

Done

L40-48: Is this a description of the current procedures used by the insurance industry?

This is a possible procedure. The methodology depends on the peril.

L45: Should trends really be taken into account by weighting of certain time periods? I would rather attempt an explicit inclusion of such trends into the approach.

Yes, an explicit inclusion of such trends would be the optimal procedure. Weighting of recent time period could be an intermediate methodology prior to major model updates.

L95: NAO

Done

L129 Public

Done

I would like to thank both reviewers for the in-depth comments, corrections and suggestions that have greatly improved the manuscript.

Please find my response below. I have answered each point individually (highlighted in blue and bold).

Reviewer 2:

The Paper by Anja Rädler deals with a very interesting and relevant topic.

Reinsurers are affected by extreme events in a relevant part of their core business, i.e. covering the risks of natural disasters.

They need to know these risks as best as possible to be able to calculate risk adequate premiums.

Climate change increasingly changes such risks. Anja Rädler describes very well in the paper how the reinsurers deal with this problem.

She also describes how changes of risks caused by natural cycles like ENSO also go into such model adaptation processes.

Anja Rädler also describes that this kind of research of climate change effects is not a one-way road as the scientists working in the insurance industry also provide additional knowledge to the rest of the scientific community, like with her project on changes in the risks of convective events.

Specific technical comments:

Some language editing is still necessary

Lines 4 and 5: The sentences should be changed in "Other drivers are changes in the frequency or intensity of natural hazards in the recent past and in the near future. In this contribution the focus will be on the latter driver."

Done

Line 34: here also already (is only done later in the paper) some reference should be made to the most recent AR6.

Yes. Done

Line 32: Exposure of "infrastructure" should be also mentioned.

Done

Paragraph around line 45: an example of natural variability should be given here.

Done

Paragraph around line 90: This should be rewritten as it is very unclear now.

Thank you for pointing this out. I have rephrased the paragraph.

Lines 110-111: There should be some discussion what a precise forecast of flood risks means for the insurability. One precondition for insurability is that a loss event cannot be foreseen in terms of time and place.

Yes, you are right. The precondition that a loss event cannot be foreseen in terms of time and place to be insurable should be discussed as well.

Line 128: …in order to be able to substantiate…

Done

Line 129: pubic -> public

Done

Line 139: …trends of the frequency of convective hazards have been…

Done